# Limitations on Variance-Reduction and Acceleration Schemes for Finite Sum Optimization

**Yossi Arjevani**
Department of Computer Science and Applied Mathematics
Weizmann Institute of Science
Rehovot 7610001, Israel
`yossi.arjevani@weizmann.ac.il`

## Abstract

We study the conditions under which one is able to efficiently apply variance-reduction and acceleration schemes on finite sum optimization problems. First, we show that, perhaps surprisingly, the finite sum structure by itself, is not sufficient for obtaining a complexity bound of $\tilde{\mathcal{O}}((n + L/\mu)\ln(1/\epsilon))$ for $L$-smooth and $\mu$-strongly convex individual functions - one must also know which individual function is being referred to by the oracle at each iteration. Next, we show that for a broad class of first-order and coordinate-descent finite sum algorithms (including, e.g., SDCA, SVRG, SAG), it is not possible to get an 'accelerated' complexity bound of $\tilde{\mathcal{O}}((n + \sqrt{nL/\mu})\ln(1/\epsilon))$, unless the strong convexity parameter is given explicitly. Lastly, we show that when this class of algorithms is used for minimizing $L$-smooth and convex finite sums, the iteration complexity is bounded from below by $\Omega(n + L/\epsilon)$, assuming that (on average) the same update rule is used in any iteration, and $\Omega(n + \sqrt{nL/\epsilon})$ otherwise.

## 1 Introduction

An optimization problem principal to machine learning and statistics is that of finite sums:

$$\min_{\mathbf{w} \in \mathbb{R}^d} F(\mathbf{w}) \coloneqq \frac{1}{n} \sum_{i=1}^n f_i(\mathbf{w}), \tag{1}$$

where the individual functions $f_i$ are assumed to possess some favorable analytical properties, such as Lipschitz-continuity, smoothness or strong convexity (see [16] for details). We measure the *iteration complexity* of a given optimization algorithm by determining how many evaluations of individual functions (via some external oracle procedure, along with their gradient, Hessian, etc.) are needed in order to obtain an $\epsilon$-solution, i.e., a point $\mathbf{w} \in \mathbb{R}^d$ which satisfies $\mathbb{E}[F(\mathbf{w}) - \min_{\mathbf{w} \in \mathbb{R}^d} F(\mathbf{w})] < \epsilon$ (where the expectation is taken w.r.t. the algorithm and the oracle randomness).

Arguably, the simplest way of minimizing finite sum problems is by using optimization algorithms for general optimization problems. For concreteness of the following discussion, let us assume for the moment that the individual functions are $L$-smooth and $\mu$-strongly convex. In this case, by applying vanilla Gradient Descent (GD) or Accelerated Gradient Descent (AGD, [16]), one obtains iteration complexity of

$$\tilde{\mathcal{O}}(n\kappa \ln(1/\epsilon)) \text{ or } \tilde{\mathcal{O}}\big(n\sqrt{\kappa}\ln(1/\epsilon)\big), \tag{2}$$

respectively, where $\kappa \coloneqq L/\mu$ denotes the *condition number* of the problem and $\tilde{\mathcal{O}}$ hides logarithmic factors in the problem parameters. However, whereas such bounds enjoy logarithmic dependence on

the accuracy level, the multiplicative dependence on $n$ renders this approach unsuitable for modern applications where $n$ is very large.

A different approach to tackle a finite sum problem is by reformulating it as a stochastic optimization problem, i.e., $\min_{\mathbf{w} \in \mathbb{R}^d} \mathbb{E}_{i \sim \mathcal{U}([n])}[f_i(\mathbf{w})]$, and then applying a general stochastic method, such as SGD, which allows iteration complexity of $\mathcal{O}(1/\epsilon)$ or $\mathcal{O}(1/\epsilon^2)$ (depending on the problem parameters). These methods offer rates which do not depend on $n$, and are therefore attractive for situations where one seeks for a solution of relatively low accuracy. An evident drawback of these methods is their broad applicability for stochastic optimization problems, which may conflict with the goal of efficiently exploiting the unique noise structure of finite sums (indeed, in the general stochastic setting, these rates cannot be improved, e.g., [1, 18]).

In recent years, a major breakthrough was made when stochastic methods specialized in finite sums (first SAG [19] and SDCA [21], and then SAGA [10], SVRG [11], SDCA without duality [20], and others) were shown to obtain iteration complexity of

$$\tilde{\mathcal{O}}((n + \kappa) \ln(1/\epsilon)). \tag{3}$$

The ability of these algorithms to enjoy both logarithmic dependence on the accuracy parameter and an additive dependence on $n$ is widely attributed to the fact that the noise of finite sum problems distributes over a finite set of size $n$. Perhaps surprisingly, in this paper we show that another key ingredient is crucial, namely, a mean of knowing which individual function is being referred to by the oracle at each iteration. In particular, this shows that variance-reduction mechanisms (see, e.g., [10, Section 3]) cannot be applied without explicitly knowing the 'identity' of the individual functions. On the more practical side, this result shows that when data augmentation (e.g., [14]) is done without an explicit enumeration of the added samples, it is impossible to obtain iteration complexity as stated in (3, see [7] for relevant upper bounds).

Although variance-reduction mechanisms are essential for obtaining an additive dependence on $n$ (as shown in (3)), they do not necessarily yield 'accelerated' rates which depend on the square root of the condition number (as shown in (2) for AGD). Recently, generic acceleration schemes were used by [13] and accelerated SDCA [22] to obtain iteration complexity of

$$\tilde{\mathcal{O}}\left((n + \sqrt{nk}) \ln(1/\epsilon)\right). \tag{4}$$

The question of whether this rate is optimal was answered affirmatively by [23, 12, 5, 3]. The first category of lower bounds exploits the degree of freedom offered by a $d$- (or an infinite-) dimensional space to show that any first-order and a certain class of second-order methods cannot obtain better rates than (4) in the regime where the number of iterations is less than $\mathcal{O}(d/n)$. The second category of lower bounds is based on maintaining the complexity of the functional form of the iterates, thereby establishing bounds for first-order and coordinate-descent algorithms whose step sizes are *oblivious* to the problem parameters (e.g., SAG, SAGA, SVRG, SDCA, SDCA without duality) for *any* number of iterations, regardless of $d$ and $n$.

In this work, we further extend the theory of oblivious finite sum algorithms, by showing that if a first-order and a coordinate-descent oracle are used, then acceleration is not possible without an explicit knowledge of the strong convexity parameter. This implies that in cases where only poor estimation of the strong convexity is available, faster rates may be obtained through 'adaptive' algorithms (see relevant discussions in [19, 4]).

Next, we show that in the smooth and convex case, oblivious finite sum algorithms which, on average, apply the same update rule at each iteration (e.g., SAG, SDCA, SVRG, SVRG++ [2], and typically, other algorithms with a variance-reduction mechanism as described in [10, Section 3]), are bound to iteration complexity of $\Omega(n + L/\epsilon)$, where $L$ denotes the smoothness parameter (rather than $\Omega(n + \sqrt{nL/\epsilon})$). To show this, we employ a restarting scheme (see [4]) which explicitly introduces the strong convexity parameter into algorithms that are designed for smooth and convex functions. Finally, we use this scheme to establish a tight dimension-free lower bound for smooth and convex finite sums which holds for oblivious algorithms with a first-order and a coordinate-descent oracle.

To summarize, our contributions (in order of appearance) are the following:

- In Section 2, we prove that in the setting of stochastic optimization, having finitely supported noise (as in finite sum problems) is not sufficient for obtaining linear convergence rates with

a linear dependence on $n$ - one must also know exactly which individual function is being referred to by the oracle at each iteration. Deriving similar results for various settings, we show that SDCA, accelerated SDCA, SAG, SAGA, SVRG, SVRG++ and other finite sum algorithms must have a proper enumeration of the individual functions in order to obtain their stated convergence rate.

- In Section 3.1, we lay the foundations of the framework of general CLI algorithms (see [3]), which enables us to formally address oblivious algorithms (e.g., when step sizes are scheduled regardless of the function at hand). In section 3.2, we improve upon [4], by showing that (in this generalized framework) the optimal iteration complexity of oblivious, deterministic or stochastic, finite sum algorithms with *both* first-order and coordinate-descent oracles cannot perform better than $\Omega(n + \kappa \ln(1/\epsilon))$, unless the strong convexity parameter is provided explicitly. In particular, the richer expressiveness power of this framework allows addressing incremental gradient methods, such as Incremental Gradient Descent [6] and Incremental Aggregated Gradient [8, IAG].

- In Section 3.3, we show that, in the $L$-smooth and convex case, the optimal complexity bound (in terms of the accuracy parameter) of oblivious algorithms whose update rules are (on average) fixed for any iteration is $\Omega(n + L/\epsilon)$ (rather then $\tilde{\mathcal{O}}(n + \sqrt{nL/\epsilon})$, as obtained, e.g., by accelerated SDCA). To show this, we first invoke a restarting scheme (used by [4]) to explicitly introduce strong convexity into algorithms for finite sums with smooth and convex individuals, and then apply the result derived in Section 3.2.

- In Section 3.4, we use the reduction introduced in Section 3.3, to show that the optimal iteration complexity of minimizing $L$-smooth and convex finite sums using oblivious algorithms equipped with a first-order and a coordinate-descent oracle is $\Omega\left(n + \sqrt{nL/\epsilon}\right)$.

## 2   The Importance of Individual Identity

In the following, we address the stochastic setting of finite sum problems (1) where one is equipped with a *stochastic* oracle which, upon receiving a call, returns some individual function chosen uniformly at random and hides its index. We show that not knowing the identity of the function returned by the oracle (as opposed to an *incremental* oracle which addresses the specific individual functions chosen by the user), significantly harms the optimal attainable performance. To this end, we reduce the statistical problem of estimating the bias of a noisy coin into that of optimizing finite sums. This reduction (presented below) makes an extensive use of elementary definitions and tools from information theory, all of which can be found in [9].

First, given $n \in \mathbb{N}$, we define the following finite sum problem

$$F_\sigma := \frac{1}{n}\left(\frac{n - \sigma}{2}f^+ + \frac{n + \sigma}{2}f^-\right), \tag{5}$$

where $n$ is w.l.o.g. assumed to be odd, $\sigma \in \{-1, 1\}$ and $f^+, f^-$ are some functions (to be defined later). We then define the following discrepancy measure between $F_1$ and $F_{-1}$ for different values of $n$ (see also [1]),

$$\delta(n) = \min_{\mathbf{w} \in \mathbb{R}^d}\{F_1(\mathbf{w}) + F_{-1}(\mathbf{w}) - F_1^* - F_{-1}^*\}, \tag{6}$$

where $F_\sigma^* := \inf_{\mathbf{w}} F_\sigma(\mathbf{w})$. It is easy to verify that no solution can be $\delta(n)/4$-optimal for both $F_1$ and $F_{-1}$, at the same time. Thus, by running a given optimization algorithm long enough to obtain $\delta(n)/4$-solution w.h.p., we can deduce the value of $\sigma$. Also, note that, one can simplify the computation of $\delta(n)$ by choosing convex $f^+, f^-$ such that $f^+(\mathbf{w}) = f^-(-\mathbf{w})$. Indeed, in this case, we have $F_1(\mathbf{w}) = F_{-1}(-\mathbf{w})$ (in particular, $F_1^* = F_{-1}^*$), and since $F_1(\mathbf{w}) + F_{-1}(\mathbf{w}) - F_1^* - F_{-1}^*$ is convex, it must attain its minimum at $\mathbf{w} = 0$, which yields

$$\delta(n) = 2(F_1(\mathbf{0}) - F_1^*). \tag{7}$$

Next, we let $\sigma \in \{-1, 1\}$ be drawn uniformly at random, and then use the given optimization algorithm to estimate the bias of a random variable $X$ which, conditioned on $\sigma$, takes $+1$ w.p. $1/2 + \sigma/2n$, and $-1$ w.p. $1/2 - \sigma/2n$. To implement the stochastic oracle described above,

conditioned on $\sigma$, we draw $k$ i.i.d. copies of $X$, denoted by $X_1, \ldots, X_k$, and return $f^-$, if $X_i = \sigma$, and $f^+$, otherwise. Now, if $k$ is such that

$$\mathbb{E}[F_\sigma(\mathbf{w}^{(k)}) - F_\sigma^* \,|\, \sigma] \leq \frac{\delta(n)}{40},$$

for both $\sigma = -1$ and $\sigma = 1$, then by Markov inequality, we have that

$$\mathbb{P}\left(F_\sigma(\mathbf{w}^{(k)}) - F_\sigma^* \geq \delta(n)/4 \,\Big|\, \sigma\right) \leq 1/10 \tag{8}$$

(note that $F_\sigma(\mathbf{w}^{(k)}) - F_\sigma^*$ is a non-negative random variable). We may now try to guess the value of $\sigma$ using the following estimator

$$\hat{\sigma}(\mathbf{w}^{(k)}) = \underset{\sigma' \in \{-1,1\}}{\operatorname{argmin}} \{F_{\sigma'}(\mathbf{w}^{(k)}) - F_{\sigma'}^*\},$$

whose probability of error, as follows by Inequality (8), is

$$\mathbb{P}\left(\hat{\sigma} \neq \sigma\right) \leq 1/10. \tag{9}$$

Lastly, we show that the existence of an estimator for $\sigma$ with high probability of success implies that $k = \Omega(n^2)$. To this end, note that the corresponding conditional dependence structure of this probabilistic setting can be modeled as follows: $\sigma \to X_1, \ldots, X_k \to \hat{\sigma}$. Thus, we have

$$H(\sigma \,|\, X_1, \ldots, X_k) \overset{(a)}{\leq} H(\sigma \,|\, \hat{\sigma}) \overset{(b)}{\leq} H_b(\mathbb{P}\,(\hat{\sigma} \neq \sigma)) \overset{(c)}{\leq} \frac{1}{2}, \tag{10}$$

where $H(\cdot)$ and $H_b(\cdot)$ denote the Shannon entropy function and the binary entropy function, respectively, $(a)$ follows by the data processing inequality (in terms of entropy), $(b)$ follows by Fano's inequality and $(c)$ follows from Equation (9). Applying standard entropy identities, we get

$$
\begin{aligned}
H(\sigma \,|\, X_1, \ldots, X_k) &\overset{(d)}{=} H(X_1, \ldots, X_k \,|\, \sigma) + H(\sigma) - H(X_1, \ldots, X_k) \\
&\overset{(e)}{=} kH(X_1 \,|\, \sigma) + 1 - H(X_1, \ldots, X_k) \\
&\overset{(f)}{\geq} kH(X_1 \,|\, \sigma) + 1 - kH(X_1),
\end{aligned} \tag{11}
$$

where $(d)$ follows from Bayes rule, $(e)$ follows by the fact that $X_i$, conditioned on $\sigma$, are i.i.d. and $(f)$ follows from the chain rule and the fact that conditioning reduces entropy. Combining this with Inequality (10) and rearranging, we have

$$k \geq \frac{1}{2(H(X_1) - H(X_1 \,|\, \sigma))} \geq \frac{1}{2\,(1/n)^2} = \frac{n^2}{2},$$

where the last inequality follows from the fact that $H(X_1) = 1$ and the following estimation for the binary entropy function: $H_b(p) \geq 1 - 4\,(p - 1/2)^2$ (see Lemma 2, Appendix A). Thus, we arrive at the following statement.

**Lemma 1.** *The minimal number of stochastic oracle calls required to obtain $\delta(n)/40$-optimal solution for problem (5) is $\geq n^2/2$.*

Instantiating this schemes for $f^+, f^-$ of various analytical properties yields the following.

**Theorem 1.** *When solving a finite sum problem (defined in 1) with a stochastic oracle, one needs at least $n^2/2$ oracle calls in order to obtain an accuracy level of:*

1. *$\frac{\kappa+1}{40n^2}$ for smooth and strongly convex individuals with condition $\kappa$.*

2. *$\frac{L}{40n^2}$ for $L$-smooth and convex individuals.*

3. *$\frac{M^2}{40\lambda n^2}$ if $\frac{M}{\lambda n} \leq 1$, and $\frac{M}{20n} - \frac{\lambda}{40}$, otherwise, for $(M+\lambda)$-Lipschitz continuous and $\lambda$-strongly convex individuals.*

**Proof**

1. Define,

$$f^{\pm}(\mathbf{w}) = \frac{1}{2}\left(\mathbf{w} \pm \mathbf{q}\right)^{\top} A \left(\mathbf{w} \pm \mathbf{q}\right),$$

where $A$ is a $d \times d$ diagonal matrix whose diagonal entries are $\kappa, 1 \ldots, 1$, and $\mathbf{q} = (1, 1, 0, \ldots, 0)^{\top}$ is a $d$-dimensional vector. One can easily verify that $f^{\pm}$ are smooth and strongly convex functions with condition number $\kappa$, and that

$$F_{\sigma}(\mathbf{w}) = \frac{1}{2}\left(\mathbf{w} - \frac{\sigma}{n}\mathbf{q}\right)^{\top} A \left(\mathbf{w} - \frac{\sigma}{n}\mathbf{q}\right) + \frac{1}{2}\left(1 - \frac{1}{n^2}\right)\mathbf{q}^{\top}A\mathbf{q}.$$

Therefore, the minimizer of $F_{\sigma}$ is $(\sigma/n)\mathbf{q}$, and using Equation (7), we see that $\delta(n) = \frac{\kappa+1}{n^2}$.

2. We define

$$f^{\pm}(\mathbf{w}) = \frac{L}{2}\left\|\mathbf{w} \pm \mathbf{e}_1\right\|^2.$$

One can easily verify that $f^{\pm}$ are $L$-smooth and convex functions, and that the minimizer of $F_{\sigma}$ is $(\sigma/n)\mathbf{e}_1$. By Equation (7), we get $\delta(n) = \frac{L}{n^2}$.

3. We define

$$f^{\pm}(\mathbf{w}) = M\|\mathbf{w} \pm \mathbf{e}_1\| + \frac{\lambda}{2}\left\|\mathbf{w}\right\|^2,$$

over the unit ball. Clearly, $f^{\pm}$ are $(M + \lambda)$-Lipschitz continuous and $\lambda$-strongly convex functions. It can be verified that the minimizer of $F_{\sigma}$ is $(\sigma \min\{\frac{M}{\lambda n}, 1\})\mathbf{e}_1$. Therefore, by Equation (7), we see that in this case we have

$$\delta(n) = \begin{cases} \frac{M^2}{\lambda n^2} & \frac{M}{\lambda n} \le 1 \\ \frac{2M}{n} - \lambda & \text{o.w.} \end{cases}.$$

∎

A few conclusions can be readily made from Theorem 1. First, if a given optimization algorithm obtains an iteration complexity of an order of $c(n, \kappa) \ln(1/\epsilon)$, up to logarithmic factors (including the norm of the minimizer which, in our construction, is of an order of $1/n$ and coupled with the accuracy parameter), for solving smooth and strongly convex finite sum problems with a stochastic oracle, then

$$c(n, \kappa) = \tilde{\Omega}\left(\frac{n^2}{\ln(n^2/(\kappa+1))}\right).$$

Thus, the following holds for optimization of finite sums with smooth and strongly convex individuals.

**Corollary 1.** *In order to obtain linear convergence rate with linear dependence on $n$, one must know the index of the individual function addressed by the oracle.*

This implies that variance-reduction methods such as, SAG, SAGA, SDCA and SVRG (possibly combining with acceleration schemes), which exhibit linear dependence on $n$, cannot be applied when data augmentation is used. In general, this conclusion also holds for cases when one applies general first-order optimization algorithms, such as AGD, on finite sums, as this typically results in a linear dependence on $n$. Secondly, if a given optimization algorithm obtains an iteration complexity of an order of $n + L^{\beta}\|\mathbf{w}^{(0)} - \mathbf{w}^*\|^2/\epsilon^{\alpha}$ for solving smooth and convex finite sum problems with a stochastic oracle, then $n + L^{\beta-\alpha}n^{2(\alpha-1)} = \Omega(n^2)$. Therefore, $\beta = \alpha$ and $\alpha \ge 2$, indicating that an iteration complexity of an order of $n + L\|\mathbf{w}^{(0)} - \mathbf{w}^*\|^2/\epsilon$, as obtained by, e.g., SVRG++, is not attainable with a stochastic oracle. Similar reasoning based on the Lipschitz and strongly convex case in Theorem 1 shows that the iteration complexity guaranteed by accelerated SDCA is also not attainable in this setting.

# 3 Oblivious Optimization Algorithms

In the previous section, we discussed different situations under which variance-reduction schemes are not applicable. Now, we turn to study under what conditions can one apply acceleration schemes. First, we define the framework of oblivious CLI algorithms. Next, we show that, for this family of algorithms, knowing the strong convexity parameter is crucial for obtaining accelerated rates. We then describe a restarting scheme through which we establish that *stationary* algorithms (whose update rule are, on average, the same for every iteration) for smooth and convex functions are sub-optimal. Finally, we use this reduction to derive a tight lower bound for smooth and convex finite sums on the iteration complexity of any oblivious algorithm (not just stationary).

## 3.1 Framework

In the sequel, following [3], we present the analytic framework through which we derive iteration complexity bounds. This, perhaps pedantic, formulation will allows us to study somewhat subtle distinctions between optimization algorithms. First, we give a rigorous definition for a *class of optimization problems* which emphasizes the role of prior knowledge in optimization.

**Definition 1** (Class of Optimization Problems). *A class of optimization problems is an ordered triple* $(\mathcal{F}, \mathcal{I}, O_f)$, *where* $\mathcal{F}$ *is a family of functions defined over some domain designated by* $dom(\mathcal{F})$, $\mathcal{I}$ *is the side-information given prior to the optimization process and* $O_f$ *is a suitable oracle procedure which upon receiving* $\mathbf{w} \in dom\mathcal{F}$ *and* $\theta$ *in some parameter set* $\Theta$, *returns* $O_f(\mathbf{w}, \theta) \subseteq dom(\mathcal{F})$ *for a given* $f \in \mathcal{F}$ *(we shall omit the subscript in* $O_f$ *when* $f$ *is clear from the context).*

In finite sum problems, $\mathcal{F}$ comprises of functions as defined in (1); the side-information may contain the smoothness parameter $L$, the strong convexity parameter $\mu$ and the number of individual functions $n$; and the oracle may allow one to query about a specific individual function (as in the case of incremental oracle, and as opposed to the stochastic oracle discussed in Section 2). We now turn to define CLI optimization algorithms (see [3] for a more comprehensive discussion).

**Definition 2** (CLI). *An optimization algorithm is called a Canonical Linear Iterative (CLI) optimization algorithm over a class of optimization problems* $(\mathcal{F}, \mathcal{I}, O_f)$, *if given an instance* $f \in \mathcal{F}$ *and initialization points* $\{\mathbf{w}_i^{(0)}\}_{i \in \mathcal{J}} \subseteq dom(\mathcal{F})$, *where* $\mathcal{J}$ *is some index set, it operates by iteratively generating points such that for any* $i \in \mathcal{J}$,

$$\mathbf{w}_i^{(k+1)} \in \sum_{j \in \mathcal{J}} O_f\left(\mathbf{w}_j^{(k)}; \theta_{ij}^{(k)}\right), \quad k = 0, 1, \dots \tag{12}$$

*holds, where* $\theta_{ij}^{(k)} \in \Theta$ *are parameters chosen, stochastically or deterministically, by the algorithm, possibly based on the side-information. If the parameters do not depend on previously acquired oracle answers, we say that the given algorithm is* oblivious. *For notational convenience, we assume that the solution returned by the algorithm is stored in* $\mathbf{w}_1^{(k)}$.

Throughout the rest of the paper, we shall be interested in oblivious CLI algorithms (for brevity, we usually omit the 'CLI' qualifier) equipped with the following two incremental oracles:

Generalized first-order oracle: $\quad O(\mathbf{w}; A, B, \mathbf{c}, i) \coloneqq A\nabla f_i(\mathbf{w}) + B\mathbf{w} + \mathbf{c}$,

Steepest coordinate-descent oracle: $\quad O(\mathbf{w}; j, i) \coloneqq \mathbf{w} + t^* \mathbf{e}_j$, $\qquad$ (13)

where $A, B \in \mathbb{R}^{d \times d}, \mathbf{c} \in \mathbb{R}^d, i \in [n], j \in [d]$, $\mathbf{e}_j$ denotes the $j$'th $d$-dimensional unit vector and $t^* \in \mathrm{argmin}_{t \in \mathbb{R}} f_j(w_1, \dots, w_{j-1}, w_j + t, w_{j+1}, \dots, w_d)$. We restrict the oracle parameters such that only one individual function is allowed to be accessed at each iteration. We remark that the family of oblivious algorithms with a first-order and a coordinate-descent oracle is wide and subsumes SAG, SAGA, SDCA, SDCA without duality, SVRG, SVRG++ to name a few. Also, note that coordinate-descent steps w.r.t. partial gradients can be implemented using the generalized first-order oracle by setting $A$ to be some principal minor of the unit matrix (see, e.g., RDCM in [15]). Further, similarly to [3], we allow both first-order and coordinate-descent oracles to be used during the same optimization process.

## 3.2 No Strong Convexity Parameter, No Acceleration for Finite Sum Problems

Having described our analytic approach, we now turn to present some concrete applications. Below, we show that in the absence of a good estimation for the strong convexity parameter, the optimal

iteration complexity of oblivious algorithms is $\Omega(n + k \ln(1/\epsilon))$. Our proof is based on the technique used in [3, 4] (see [3, Section 2.3] for a brief introduction of the technique).

Given $0 < \epsilon < L$, we define the following set of optimization problems (over $\mathbb{R}^d$ with $d > 1$)

$$F_\mu(\mathbf{w}) := \frac{1}{n} \sum_{i=1}^{n} \left( \frac{1}{2} \mathbf{w}^\top Q_\mu \mathbf{w} - \mathbf{q}^\top \mathbf{w} \right), \text{ where} \tag{14}$$

$$Q_\mu := \begin{pmatrix} \frac{L+\mu}{2} & \frac{\mu-L}{2} & & & \\ \frac{\mu-L}{2} & \frac{L+\mu}{2} & & & \\ & & \mu & & \\ & & & \ddots & \\ & & & & \mu \end{pmatrix}, \quad \mathbf{q} := \frac{\epsilon R}{\sqrt{2}} \begin{pmatrix} 1 \\ 1 \\ 0 \\ \vdots \\ 0 \end{pmatrix},$$

parametrized by $\mu \in (\epsilon, L)$ (note that the individual functions are identical. We elaborate more on this below). It can be easily verified that the condition number of $F_\mu$, which we denote by $\kappa(F_\mu)$, is $L/\mu$, and that the corresponding minimizers are $\mathbf{w}^*(\mu) = (\epsilon R/\mu\sqrt{2}, \epsilon R/\mu\sqrt{2}, 0, \ldots, 0)^\top$ with norm $\leq R$.

If we are allowed to use different optimization algorithm for different $\mu$ in this setting, then we know that the optimal iteration complexity is of an order of $(n + \sqrt{n\kappa(F_\mu)}) \ln(1/\epsilon)$. However, if we allowed to use only one single algorithm, then we show that the optimal iteration complexity is of an order of $n + \kappa(F_\mu) \ln(1/\epsilon)$. The proof goes as follows. First, note that in this setting, the oracles defined in (13) take the following form,

Generalized first-order oracle: $O(\mathbf{w}; A, B, \mathbf{c}, i) = A(Q_\mu \mathbf{w} - \mathbf{q}) + B\mathbf{w} + \mathbf{c}$, (15)

Steepest coordinate-descent oracle: $O(\mathbf{w}; j, i) = (I - (1/(Q_\mu)_{jj})\mathbf{e}_i(Q_\mu)_{j,*}) \mathbf{w} - q_j/(Q_\mu)_{jj}\mathbf{e}_j$.

Now, since the oracle answers are linear in $\mu$ and the $k$'th iterate is a $k$-fold composition of sums of the oracle answers, it follows that $\mathbf{w}_1^{(k)}$ forms a $d$-dimensional vector of univariate polynomials in $\mu$ of degree $\leq k$ with (possibly random) coefficients (formally, see Lemma 3, Appendix A). Denoting the polynomial of the first coordinate of $\mathbb{E}\mathbf{w}_1^{(k)}(\mu)$ by $s(\mu)$, we see that for any $\mu \in (\epsilon, L)$,

$$\mathbb{E}\|\mathbf{w}_1^{(k)}(\mu) - \mathbf{w}^*(\mu)\| \geq \|\mathbb{E}\mathbf{w}_1^{(k)}(\mu) - \mathbf{w}^*(\mu)\| \geq \left| s(\mu) - \frac{R\epsilon}{\sqrt{2}\mu} \right| \geq \frac{R\epsilon}{\sqrt{2}L} \left| \frac{\sqrt{2}s(\mu)\mu}{R\epsilon} - 1 \right|,$$

where the first inequality follows by Jensen inequality and the second inequality by focusing on the first coordinate of $\mathbb{E}\mathbf{w}^{(k)}(\eta)$ and $\mathbf{w}^*(\eta)$. Lastly, since the coefficients of $s(\mu)$ do not depend on $\mu$, we have by Lemma 4 in Appendix A, that there exists $\delta > 0$, such that for any $\mu \in (L - \delta, L)$ it holds that

$$\frac{R\epsilon}{\sqrt{2}L} \left| \frac{\sqrt{2}s(\mu)\mu}{R\epsilon} - 1 \right| \geq \frac{R\epsilon}{\sqrt{2}L} \left( 1 - \frac{1}{\kappa(F_\mu)} \right)^{k+1},$$

by which we derive the following.

**Theorem 2.** *The iteration complexity of oblivious finite sum optimization algorithms equipped with a first-order and a coordinate-descent oracle whose side-information does not contain the strong convexity parameter is $\tilde{\Omega}(n + \kappa \ln(1/\epsilon))$.*

The $n$ part of the lower bound holds for any type of finite sum algorithm and is proved in [3, Theorem 5]. The lower bound stated in Theorem 2 is tight up to logarithmic factors and is attained by, e.g., SAG [19]. Although relying on a finite sum with identical individual functions may seem somewhat disappointing, it suggests that some variance-reduction schemes can only give optimal dependence in terms of $n$, and that obtaining optimal dependence in terms of the condition number need to be done through other (acceleration) mechanisms (e.g., [13]). Lastly, note that, this bound holds for any number of iterations (regardless of the problem parameters).

### 3.3 Stationary Algorithms for Smooth and Convex Finite Sums are Sub-optimal

In the previous section, we showed that not knowing the strong convexity parameter reduces the optimal attainable iteration complexity. In this section, we use this result to show that whereas general

optimization algorithms for smooth and convex finite sum problems obtain iteration complexity of $\tilde{\mathcal{O}}(n + \sqrt{nL/\epsilon})$, the optimal iteration complexity of stationary algorithms (whose expected update rules are fixed) is $\Omega(n + L/\epsilon)$.

The proof (presented below) is based on a general restarting scheme (see Scheme 1) used in [4]. The scheme allows one to apply algorithms which are designed for $L$-smooth and convex problems on smooth and strongly convex finite sums by explicitly incorporating the strong convexity parameter. The key feature of this reduction is its ability to 'preserve' the exponent of the iteration complexity from an order of $C(f)(L/\epsilon)^\alpha$ in the non-strongly convex case to an order of $(C(f)\kappa)^\alpha \ln(1/\epsilon)$ in the strongly convex case, where $C(f)$ denotes some quantity which may depend on $f$ but not on $k$, and $\alpha$ is some positive constant.

| SCHEME 1 | RESTARTING SCHEME |
|---|---|
| GIVEN | AN OPTIMIZATION ALGORITHM $\mathcal{A}$ |
| | FOR SMOOTH CONVEX FUNCTIONS WITH |
| | $f(\mathbf{w}^{(k)}) - f^* \leq \frac{C(f)\left\|\bar{\mathbf{w}}^{(0)} - \mathbf{w}^*\right\|^2}{k^\alpha}$ |
| | FOR ANY INITIALIZATION POINT $\bar{\mathbf{w}}^0$ |
| ITERATE | FOR $t = 1, 2, \ldots$ |
| | RESTART THE STEP SIZE SCHEDULE OF $\mathcal{A}$ |
| | INITIALIZE $\mathcal{A}$ AT $\bar{\mathbf{w}}^{(0)}$ |
| | RUN $\mathcal{A}$ FOR $\sqrt[\alpha]{4C(f)/\mu}$ ITERATIONS |
| | SET $\bar{\mathbf{w}}^{(0)}$ TO BE THE POINT RETURNED BY $\mathcal{A}$ |
| END | |

The proof goes as follows. Suppose $\mathcal{A}$ is a stationary CLI optimization algorithm for $L$-smooth and convex finite sum problems equipped with oracles (13). Also, assume that its convergence rate for $k \geq N$, $N \in \mathbb{N}$ is of an order of $\frac{n^\gamma L^\beta \left\|\mathbf{w}^{(0)} - \mathbf{w}^*\right\|^2}{k^\alpha}$, for some $\alpha, \beta, \gamma > 0$. First, observe that in this case we must have $\beta = 1$. For otherwise, we get $f(\mathbf{w}^{(k)}) - f^* = ((\nu f)(\mathbf{w}^{(k)}) - (\nu f)^*)/\nu \leq n^\gamma (\nu L)^\beta/\nu k^\alpha = \nu^{\beta-1} n^\gamma L^\beta/k^\alpha$, implying that, simply by scaling $f$, one can optimize to any level of accuracy using at most $N$ iterations, which contradicts [3, Theorem 5]. Now, by [4, Lemma 1], Scheme 1 produces a new algorithm whose iteration complexity for smooth and strongly convex finite sums with condition number $\kappa$ is

$$\mathcal{O}(N + n^\gamma (L/\epsilon)^\alpha) \longrightarrow \tilde{\mathcal{O}}(n + n^\gamma \kappa^\alpha \ln(1/\epsilon)). \tag{16}$$

Finally, stationary algorithms are invariant under this restarting scheme. Therefore, the new algorithm cannot depend on $\mu$. Thus, by Theorem 2, it must hold that that $\alpha \geq 1$ and that $\max\{N, n^\gamma\} = \Omega(n)$, proving the following.

**Theorem 3.** *If the iteration complexity of a stationary optimization algorithm for smooth and convex finite sum problems equipped with a first-order and a coordinate-descent oracle is of the form of the l.h.s. of (16), then it must be at least $\Omega(n + L/\epsilon)$.*

We note that, this lower bound is tight and is attained by, e.g., SDCA.

### 3.4 A Tight Lower Bound for Smooth and Convex Finite Sums

We now turn to derive a lower bound for finite sum problems with smooth and convex individual functions using the restarting scheme shown in the previous section. Note that, here we allow any oblivious optimization algorithm, not just stationary. The technique shown in Section 3.2 of reducing an optimization problem into a polynomial approximation problem was used in [3] to derive lower bounds for various settings. The smooth and convex case was proved only for $n = 1$, and a generalization for $n > 1$ seems to reduce to a non-trivial approximation problem. Here, using Scheme 1, we are able to avoid this difficulty by reducing the non-strongly case to the strongly convex case, for which a lower bound for a general $n$ is known.

The proof follows the same lines of the proof of Theorem 3. Given an oblivious optimization algorithm for finite sums with smooth and convex individuals equipped with oracles (13), we apply again Scheme 1 to get an algorithm for the smooth and strongly convex case, whose iteration complexity is as in (16). Now, crucially, oblivious algorithm are invariant under Scheme 1 (that

is, when applied on a given oblivious algorithm, Scheme 1 produces another oblivious algorithm). Therefore, using [3, Theorem 2], we obtain the following.

**Theorem 4.** *If the iteration complexity of an oblivious optimization algorithm for smooth and convex finite sum problems equipped with a first-order and a coordinate-descent oracle is of the form of the l.h.s. of (16), then it must be at least*

$$\Omega\left(n + \sqrt{\frac{nL}{\epsilon}}\right).$$

This bound is tight and is obtained by, e.g., accelerated SDCA [22]. Optimality in terms of $L$ and $\epsilon$ can be obtained simply by applying Accelerate Gradient Descent [16], or alternatively, by using an accelerated version of SVRG as presented in [17]. More generally, one can apply acceleration schemes, e.g., [13], to get an optimal dependence on $\epsilon$.

### Acknowledgments

We thank Raanan Tvizer and Maayan Maliach for several helpful and insightful discussions.

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
