[Supplementary Material]

# A    Technical Lemmas

**Lemma 2.** *Let $H_b(p) := -p \log_2 p - (1-p) \log_2(1-p)$, be the binary entropy function. Then,*

$$H_b(p) \geq 1 - 4(p - 1/2)^2.$$

**Proof** First, note that the first two derivatives of $H$ are

$$H_b'(p) = \log_2(1-p) - \log_2 p,$$

$$H_b''(p) = -\frac{1}{\ln(2)p(1-p)}.$$

We show that the following function

$$\varphi(p) := H_b(p) - \left(1 - 4\left(p - \frac{1}{2}\right)^2\right),$$

is non-negative on $[0, 1]$ (note that, since $\varphi$ is continuous, it is bounded from below on $[0, 1]$ and its minimum is attained on some local minimum in $[0, 1]$). Let us locate all the extrema points of $\varphi$ in $(0, 1)$. We have that,

$$\varphi'(p) = \log_2\left(\frac{1-p}{p}\right) + 8\left(p - \frac{1}{2}\right).$$

Therefore, $\varphi(1/2) = 0$, and since

$$\varphi''(p) = \frac{-1}{\ln(2)x(1-x)} + 8,$$

it follows that $\varphi''(1/2) > 0$, which implies that $p = 1/2$ is a local minimum of $\varphi$. We claim that there are exactly two more extrema points of $\varphi$ which are in fact local maximum points. To this end, note that

$$\varphi''(p) \begin{cases} > 0 & |p - 1/2| < c, \\ = 0 & |p - 1/2| = c, \\ < 0 & |p - 1/2| > c, \end{cases}$$

where $c := \sqrt{1/4 - 1/(8\ln(2))}$. Therefore, by Rolle's Theorem, $\varphi'$ does not vanish in $0 < |p - 1/2| \leq c$, and vanishes exactly once in $p > 1/2 + c$ and exactly once in $p < 1/2 - c$. Since, $\varphi''$ is strictly negative in $|p - 1/2| > c$, it follows that the other two stationary points of $\varphi'$ are local maxima of $\varphi$. All in all, we have that if $p' \in [0, 1]$ is a local minimum of $\varphi$, then $p' \in \{0, 1/2, 1\}$, which implies that

$$\varphi(p) \geq \min\{\varphi(0), \varphi(1/2), \varphi(1)\} = 0,$$

concluding the proof. ∎

**Lemma 3.** *When applied on problem (14) with oracles (15), the coordinates of iterates produced by oblivious stochastic CLIs form polynomials in $\mu$ with random coefficients (which do not depend on $\mu$) and whose degrees do not exceed the iteration number.*

**Proof** Let $\mathcal{A}$ be an oblivious stochastic CLI, and suppose we apply $\mathcal{A}$ on the class of problems (14) parametrized by $\mu$, using oracles (15). We use mathematical induction to show that for any $k = 0, 1, \ldots$, the coordinates of the $k$'th iterate produced by such process can be expressed as distributions over $\mathcal{P}_k$, where $\mathcal{P}_k$ denotes the set of all real polynomials with degree $\leq k$.

As the first iterate $\mathbf{w}_i^{(0)}$ is allowed to depend only on $L$ and $n$, the base case is trivial. For the inductive step, assume that any coordinate of $\mathbf{w}_i^{(k)}$ can be expressed as a distribution over $\mathcal{P}_k$. Now, for any $\mathbf{w}_i^{(k)}$, the oracles answers of

Generalized first-order oracle:

$$O(\mathbf{w}_i^{(k)}; A, B, \mathbf{c}, i) = A(Q_\mu \mathbf{w}_i^{(k)} - \mathbf{q}) + B\mathbf{w}_i^{(k)} + \mathbf{c}$$

Steepest coordinate-descent oracle:

$$O(\mathbf{w}_i^{(k)}; j, i) = (I - (1/(Q_\mu)_{jj})\mathbf{e}_i(Q_\mu)_{j,*})\mathbf{w}_i^{(k)} - q_j/(Q_\mu)_{jj}\mathbf{e}_j \tag{17}$$

form a distribution over $\mathcal{P}_{k+1}$, as the random quantities involved in the expressions ($A, B, j$ and $i$) do not depend on $\mu$ (due to obliviousness) and the rest of the terms are either constant or linear in $\mu$. Lastly, $\mathbf{w}_i^{(k+1)}$ are computed by simply summing up all the oracle answers, and as such, form again distributions over $\mathcal{P}_{k+1}$. ∎

**Lemma 4.** *Let $s(\mu)$ be a real polynomial of degree $\leq k$, and let $L > 0$. Then, there exists $\delta > 0$ such that for any $\mu \in (L - \delta, L)$ it holds that*

$$|s(\mu)\mu + 1| \geq (1 - \mu/L)^{k+1}.$$

**Proof** Assume for the sake of contradiction that for any $\delta > 0$, there exists $\mu \in (L - \delta, L)$ such that

$$|s(\mu)\mu + 1| < \left(1 - \frac{\mu}{L}\right)^{k+1}.$$

Define

$$q(\mu) := s\left(L(1 - \mu)\right) L(1 - \mu) + 1 \tag{18}$$

and denote the corresponding coefficients by $q(\mu) = \sum_{j=0}^{k+1} q_i \mu^j$. We show by induction that $q_j = 0$ for all $j = 0, \ldots, k$. For $j = 0$ we have that since for any $\delta > 0$ there exists some $\hat{\mu} \in (0, 1 - (L - \delta)/L)$ such that

$$|q(\hat{\mu})| < \left(1 - \frac{L(1 - \hat{\mu})}{L}\right)^{k+1} = \hat{\mu}^{k+1},$$

it holds, by continuity, that

$$|q_0| = |q(0)| = \left| \lim_{\mu \to 0^+} q(\mu) \right| \leq \lim_{\mu \to 0^+} \mu^{k+1} = 0.$$

Now, if $q_0 = \cdots = q_{m-1} = 0$ for $m < k + 1$ then

$$|q_m| = \left| \frac{q(0)}{\mu^m} \right| = \left| \lim_{\mu \to 0^+} \frac{q(\mu)}{\mu^m} \right| \leq \lim_{t \to 0^+} \mu^{k+1-m} = 0.$$

Thus, proving the induction claim. This, in turns, implies that $q(\mu) = q_{k+1}\mu^{k+1}$. Now, by Equation (18), it follows that $q_{k+1} = q(1) = 1$. Hence, $q(\mu) = \mu^{k+1}$. Lastly, using Equation (18) again yields

$$s(\mu)\mu + 1 = q\left(1 - \frac{\mu}{L}\right) = \left(1 - \frac{\mu}{L}\right)^{k+1},$$

which contradicts our assumption, thus concluding the proof. ∎