[Reviews · NeurIPS 2017]

Reviewer 1



This papers covers some theoretical limitations on variance-reduced stochastic algorithms for finite sum problems (SAG, SDCA, SAGA, SVRG etc.). Such algorithms have had some important impact in machine learning & convex optimization as they have been shown to reach a linear convergence for strongly convex problems with lipschitz gradient (log term) while having a factor in $n + \kappa$ (and not as $n\kappa$). A first result of this paper is that the sequential iterative algorithm needs to know which individual function has been returned by the oracle to reach the linear convergence rate. Typically SAG or SAGA need this knowledge to update the gradient memory buffer so it is not terribly surprising that knowing the function index is necessary. It is then argued that such variance-reduced methods cannot be used with data augmentation during learning. This is only true if the data augmentation is stochastic and if there is not a finite set of augmented samples. A second result concerns the possibility to accelerate in the "Nesterov sense" such variance-reduced algorithm (go from $\kappa$ to $\sqrt{\kappa}}$). It is shown that "oblivious" algorithms whose update rules are (on average) fixed for any iteration cannot be accelerated. A practical consequence of the first result in 3.2 is that knowing the value of the strong convexity parameter is necessary to obtain an accelerated convergence when the algorithm is stationary. Such oblivious algorithms are further analysed using restarts which are well known techniques to alleviate the non-adaptive nature of accelerated algorithms in the presence of unknown local conditionning. The paper is overhaul well written with some pedagogy. Minor: CLI is first defined in Def 2 but is used before. Typo: L224 First, let us we describe -> First, let us describe

Reviewer 2



This paper establishes a number of results concerning lower bounds for finite sum optimisation problems. Over the last 2 years a close to complete theory has been established, which this paper contributes further to. This is an important area of research in my opinion. This paper is very well written and is more polished than most NIPS submissions. I don't have many comments: • I'm not following the use of Fano's inequality in equation (10), could you please clarify? • Should the last part of the equation between lines 134-135 be n^2/2 instead? I'm not following it. • I can't find the definition of t^{*} in equation 13. • I think the tilde is not used consistently when big-Omega notation is used in the paper. • In definition 2, it would help if it's more clearly stated that \theta typically contains the function index. Currently you just have some text before the definition “the oracle may allow one to query about a specific individual function”. • How does the result of section 3.2 relate to Theorem 2 in reference 4? Is it just a strengthening by the additional n term? . The result of section 3.3 is also closely related to results in that paper. In the introduction you state that you improve on the results in [4], I think the exact nature of the improvement needs to be more clearly stated.